# Diagnosis, Treatment, and Prevention of HIV Infection among Detainees: A Review of the Literature

**DOI:** 10.3390/healthcare10122380

**Published:** 2022-11-27

**Authors:** Ylenia Russotto, Cristina Micali, Natascia Laganà, Andrea Marino, Edoardo Campanella, Benedetto Maurizio Celesia, Giovanni Francesco Pellicanò, Emmanuele Venanzi Rullo, Giuseppe Nunnari

**Affiliations:** 1Unit of Infectious Diseases, Department of Clinical and Experimental Medicine, University of Messina, 98124 Messina, Italy; 2Biomedical and Biotechnological Sciences Department, University of Catania, 95131 Catania, Italy; 3Unit of Infectious Diseases, Department of Clinical and Experimental Medicine, ARNAS Garibaldi Nesima Hospital, University of Catania, 95122 Catania, Italy; 4Department of Human Pathology of the Adult and the Developmental Age “G. Barresi”, University of Messina, 98124 Messina, Italy

**Keywords:** HIV, jail, AIDS, prison, STI

## Abstract

Detainees are one of the most vulnerable populations to human immunodeficiency virus (HIV). This is mostly caused by the lack of knowledge on the topic among the inmates; the lack of prophylaxis; the high percentage of risky behaviors in jail, such as sexual abuse, unprotected sexual intercourses, and injective drug use; and the generally low perception of the risk of transmission. It has also been observed that the problem does not cease to exist at the moment of release, but it also may be aggravated by the weak support system or the total absence of programs for people living with HIV/AIDS (PLWHA) to avoid discontinuation of antiretroviral drugs. Difficulty in providing housing and jobs and, therefore, a form of stability for ex-detainees, also contributes to none adherence to antiretroviral therapy. Among the detainees, there are also categories of people more susceptible to discrimination and violence and, therefore, to risky behaviors, such as black people, Hispanics, transgender people, and men who have sex with men (MSM). We reviewed the literature in order to provide a more complete picture on the situation of PLWHA in jail and to also analyze the difficulties of ex-detainees in adhering to HIV therapy.

## 1. Introduction

Among inmates, the prevalence of human immunodeficiency virus (HIV) is from 2.4 to 5-fold more than the general population [1,2,3]. Psychiatric disorders, infections, drugs use, and chronic diseases are spread among inmates [4]. HIV infection and transmission in jail is a growing public health issue in the USA, especially in the southern countries, where there are the highest rates of both HIV infection and incarceration [5]. Moreover, there also are racial disparities in the prevalence of HIV infection and detention. It is not surprising to find that half of the new HIV infections are diagnosed in black people, since they represent 42% of the jail population, with the Hispanic population being the second more frequent ethnic group in jail [1,6]. These ethnic groups are, consequently, more at risk for HIV infection [7].

Several studies show that inmates are at a higher risk HIV infection than the general population not only because of a higher prevalence of risky behaviors, such as drug abuse, but also because of the superficial knowledge about HIV infection and transmission [1,8]. 

As is known, HIV infection is diffused among drug users through injections [9,10]. The substance abuse, and in particular methamphetamine, leads to a decrease in efficacy in suppressing HIV RNA plasma viral load (pVL) [1], which is related to a higher risk of comorbidities, such as chronic kidney disease, cardiovascular disease, dyslipidemia, hepatitis B and hepatitis C, acquired immunodeficiency syndrome (AIDS)-defining cancers (ADCs), and non-ADCs (NADCs) [11,12,13,14,15,16,17,18]. Therefore, risky behaviors that are commonly diffused among detainees could increase the risk of long-term consequences that are related to permanent detectable viremia. 

The aim of our review is to investigate the conditions that favor HIV spread among detainees, the issues people living ith HIV (PLWH) have to face while being in jail, and the difficulties in maintaining compliance to ART after release. We also aim to suggest interventions that could improve the situation of PLWH in prisons and the knowledge on the topic among inmates.

## 2. Materials and Methods

This review was conducted by examining publications on HIV-infected people and the risk of infection in jail on PubMed, using the keywords “HIV + jail”, “HIV + prison”, “HIV + detainees”, “Human Immunodeficiency Virus + jail”, “Human Immunodeficiency Virus + prison”, “AIDS + jail”, “AIDS + prison”, “HIV comorbidities”, and “HIV complications” for our research.

We did not set temporal limits in selecting the articles, although we mostly found publications from 2000 to 2022. We considered only articles in English and Italian and those regarding humans.

Languages different from the ones mentioned and different species from human were our only exclusion criteria.

We found a total of 86 publications.

We also performed research about other infectious diseases and their spread in prison, using “STI + jail”, “sexually transmitted infections + jail”, “parasitic infections + jail”, “helminths + prison”, “helminths + jail” “tuberculosis + jail”, “SARS-CoV-2 + jail”, and “COVID19 + jail” as keywords.

We found 28 articles.

## 3. Risk Factors and Challenges for Retention in Care

A study by Fenomanana et al. [8] performed on HIV-negative inmates highlights an inadequate knowledge about HIV. In fact, 65.04% of the interviewees thought that HIV infection was transmitted by shaking hands, while 47.97% thought it was transmitted by mosquito bites. This limited knowledge constitutes the primary barrier to stopping the virus spread among inmates [19]. Moreover, another survey shows that the 88% of 417 inmates had never heard about pre-exposure prophylaxis (PrEP), although a high percentage of them were interested in learning about it [20]. In fact, a high percentage of HIV infections in jail involves men who have sex with men (MSM) [21,22], because jails are usually divided by sex. In fact, , in the survey by Khan et al., about 40% of the 532 HIV-positive MSM declared to have been incarcerated [21].

According to a recent study performed in New York City jails, PrEP could be an effective measure to decrease the rate of HIV infections during detention, especially in MSM [23]. This study investigated sex behaviors of MSM inmates, showing that a high percentage of them engaged in unprotected sex, and especially anal intercourse [23]. When asked about their reasons for engaging in unprotected intercourse, they answered they had previously participated in unprotected anal sex with the same person, or that sex without a condom had a more “natural” feeling, or that they were under the influence of drugs [23]. Many of them also answered that they did not want to think about HIV risk [23].

Transgender people also are a vulnerable population, especially those who did not undergo registry re-attribution of sex and are placed in jails according to their birth sex. There, they often experience sexual harassment, physical abuse, and violence [24]. Female inmates are significantly more burdened by sexual abuse than male ones, even in jail. Violent sexual intercourses are common among inmates, and a survey by Meyer et al. [25] showed that among 84 HIV-positive participants, 49% suffered from sexual abuse, with the higher percentage among women than men (73% vs. 37%). Episodes of violence are mostly associated with HIV infection, since they share some risk factors, such as drug abuse and mental disorders. Moreover, a habit of violent sexual intercourse, drug abuse, and mental disorders are related to antiretroviral therapy (ART) discontinuation and low interest in healthcare [25]. 

Screening for HIV infection is still the most important pillar to avoid its spread in jail [26]. However, there is no agreement on the most effective approach. Many prisons have chosen the “opt-out” approach, therefore, they test each inmate unless they refuse their consent [27,28]. The “opt-in” strategy, on the other hand, requires that the healthcare worker gives information about HIV transmission and AIDS and asks for the inmate consent before the test. Both the strategies have their pros and their cons. If the “opt-out” approach allows to test everyone unless clearly denied by the inmate, the “opt-in” approach gives the opportunity to educate the inmate and it probably has a long-term effect [29,30]. Moreover, the “opt-in” strategy is preferred by the inmate, since it allows them to have a choice and freely adhere to the screening [30].

Differently, in Italy it is warranted by law that the inmates have to be informed about their health and have to consent to treatment and tests [31].

According to data from HIV care cascade [32], the percentage of diagnoses in the US and Canada do not significantly differ upon incarceration and after release (78% vs. 79%), showing an effective system of screening; moreover, it is similar to the national percentage of diagnosis in the population.

Routine testing has been shown to be more cost-effective than target testing [33]. In fact, it can decrease HIV transmission rate by identifying recently infected people; therefore, these people can be promptly started on ART [33]. This would allow for avoiding the costs for therapies and medical care in the case of AIDS-related and non-AIDS-related comorbidities [33].

Major risk factors are enumerated in Table 1.

## 4. Infectious Comorbidities

According to the U.S. Preventive Services Task Force, all the inmates must be screened for HIV infection, hepatitis C virus (HCV) infection, syphilis, latent tuberculosis (LTB) infection, chlamydia (CT), and gonorrhea (GC) [4]. In fact, during detention, the risk of the acquisition of blood-borne pathogens, airborne organisms such as mycobacterium tuberculosis, severe acute respiratory syndrome coronavirus 2 (SARS-CoV-2), influenza virus, varicella-zoster virus, sexually transmitted infections (STIs), methicillin-resistant *Staphylococcus aureus* (MRSA) infection, and parasitic infection is increased [34,35,36,37].

### 4.1. Sexually Transmitted Infection

In the last guidelines for diagnosis and treatment of STIs, Workowski et al. [38] recommend a universal opt-out approach, especially in people under 30 years of age.

STIs are often difficult to diagnose because of the lack of symptoms; however, they can lead to adverse outcomes such as infertility or, in the long-term, cancer [39,40,41,42,43,44]. The prevalence of STIs among inmates is significantly higher than the general population [45,46]. A cross-sectional study on 676 prisoners in Stockholm County found that the prevalence of viremic HCV infection among Swedish inmates was 11.5%, higher than the general population [47]. Moreover, HIV infection is often accompanied by other STIs, such as syphilis, HBV, HCV, chlamydia, and gonorrhea, for they cause the appearance of mucosal lesions in the genital area, which can facilitate HIV transmission [48]. 

Screening programs in US correctional facilities, such as prisons and jails, provide an important opportunity to diagnose and treat people at high risk of being infected, who may otherwise lack the resources to obtain tests and, eventually, appropriate STI management [45]. 

In 2020, Dang CM et al. paired CT/GC testing with routine urine pregnancy tests in incarcerated women, finding an increase of 4.7-fold in the monthly STI testing rates and a comparable increase in the number of infections detected [39]. On the other hand, a study performed in a large county jail in Chicago showed that the discontinuation of male CT and GC universal screening was a missed opportunity not only to screen a high-risk population, but mostly to determine a substantial decline in morbidity [49]. 

Among inmates, the estimated global prevalence for HCV infection is 17.7%, with the highest rates in Australia and Oceania, followed by Europe [50]. The main risk factors for transmission are previous detention, ignorance of transmission routes, alcohol consumption for HIV and both HBV and HCV, intra-prison anal sex, and multiple sex partners for HIV and HBV [51]. Other risk factors are represented by tattoos during incarceration using recycled materials and the injection of drugs with needle-sharing before and during incarceration [52]. Despite the high risk of infection, HBV vaccination rate among inmates is lower than in general population, underlying the urgent necessity of vaccination program implementation in detention facilities [53].

Age of inmates is also relevant to determine the risk of acquiring an STI. In fact, people aged 21–26 years have a higher risk of being infected with HIV and HBV, while people aged 33–38 years have a higher risk of HCV infection [51]. 

The prevalence of syphilis infection among inmates is higher than in the general population [54]. Among male inmates, syphilis infection is associated with a history of genital ulcer and uncircumcised status [53]. Female inmates seem to have a higher prevalence of HIV and syphilis co-infection during pregnancy, probably due to the higher levels of social vulnerability and to the lower quality of prenatal care [55].

### 4.2. Helminths and Intestinal Parasitic Infections

Low household income, poor personal and environmental hygiene, overcrowding, limited access to clean water, tropical climate, and low altitude are elements strictly associated with the development and spread of intestinal parasitic infections (IPIs). Homeless people and inmates are often obligated to live in inadequate facilities [56]. In jail, helminths and other IPIs are very common infections, and they may cause serious life-threatening diseases in inmates of developing countries. 

A study of IPIs performed in southern Ethiopia on 400 samples showed a prevalence of 6%, mostly in male (M:F ratio 1.7:1) of about 21–40 years and in the ‘Dalits’ ethnic group. In 62.5% of the cases, protozoans were isolated, with *Giardia lamblia* being the most common (41.67%). In the rest of the cases, helminths were found, with *Trichuris trichiura* (25.0%) being the most reported. The main risk factors associated with the parasitic intestinal infections were lack of hand washing and sleeping in groups [57]. It is not uncommon to diagnose a co-infection of *Giardia lamblia* and *Entamoeba* spp. in inmates, especially in northern Ethiopia [58]. 

The implementation of an effective deworming campaign, the education on pure and safe drinking water supply, sanitation, and hygiene (WASH), trimming of fingernails, and the introduction of periodic screening for intestinal parasitic infection could effectively reduce the infestations in prison [36,57,59].

### 4.3. Tuberculosis and Latent Tuberculosis

Several studies demonstrated that inmates are at high risk of HIV and tuberculosis (TB) co-infections, with a high rate of severe morbidity in jail settings [29,60]. Nonetheless, the use of combined screening in jail is limited [29]. 

The median estimated annual TB incidence rate from 2002 to 2013 was 29 cases/100,000 for local jail inmates. This rate dropped to 8/100,000 for state inmates, and to 25/100,000 for federal inmates. TB infection in jail is prevalent in females and in US-born prisoners, according to the case reported on the National Tuberculosis Surveillance System [61]. However, according to analysis on Bureau of Justice Statistics surveys of federal and state prisons in the US, there are some racial disparities in TB/HIV screening that can contribute to health disparities in the communities with the highest rate of imprisonment [62]. TB/HIV screening rates for US-born Hispanics, foreign-born Hispanics, non-Hispanic blacks, and non-Hispanic whites were analyzed. Despite screening rates being high overall, foreign-born Hispanic inmates have significantly lower probability of being tested for TB either in state or federal prisons in comparison to white inmates detainees [62]. 

To limit the transmission of HIV and TB infections among detainees, it is necessary to apply of screening protocols at the arrest [61].

### 4.4. SARS-CoV-2 Infection

From the beginning of SARS-CoV-2 pandemic in China, the coronavirus has progressively spread worldwide, not even sparing detention facilities [63]. Fragile people such as pregnant women, elders, and those with medical comorbidities such as diabetes, cerebrovascular disorders, cardiovascular diseases, chronic pulmonary disease, and compromised immunity, are at higher risk of COVID-19 complications and adverse outcomes [64,65,66,67]. Moreover, in detention facilities, not to be housed in single-cell units, low levels of prison security, and crowding are important risk factors for SARS-CoV-2 transmission [68,69].

The pandemic also had a devastating effect on the supply chain worldwide. The consequences affect people dependent on the access to medications such as people on ART, those taking PrEP, and opioid agonist treatment (OAT) [67]. According to the July 2020 report of the World Health Organization (WHO), during the COVID-19 pandemic, provision of ART was discontinued in several countries [69]. Moreover, the pandemic badly affected services such as monitoring of viral load, HIV and STI screening, and condom and needle exchange facilities [67]. Furthermore, clinicians worldwide were reallocated to address the emergency, leaving PLWH with an uncertain access to treatment and cures [67]. 

New drugs administration routes, such as injective antiretroviral drugs, will be useful to promote adherence, even if similar situations will rise in the future, especially in detention settings [70].

Guaranteeing equal access to vaccination is also a priority, in order to decrease the spread of SARS-CoV-2 in prison [71].

## 5. ART in Jail

PLWH entering prisons are often exposed to a suboptimal healthcare system [72]. Access to a healthcare system is of utmost importance to guarantee a good adherence to ART and, therefore, to reduce the percentage of viremic people [72]. As a matter of fact, the spread of the HIV infection is also related to the number of infected inmates and their adherence to ART [2]. In the United States, 23–68% of homeless people have a history of incarceration and, unfortunately, they do not have a facilitated access to healthcare out of jail; therefore, their compliance to ART is suboptimal. [72]. In addition, upon incarceration, some known HIV-positive individuals are not on ART nor have undetectable serum HIV viremia [2]. Chen et al. [72] compared ART adherence between homeless and not-homeless inmates, and they showed that this inadequate adherence poses a high risk for HIV infection spread in jail. On the other hand, ART is generally available for HIV-positive inmates. Eastment et al. [73] showed that although 51% of the HIV-positive inmates had a detectable pVL upon their arrest, 62% of them had still an undetectable pVL within a year of release. Interestingly, according to data from HIV care cascade, the peaks in linkage and retention in care (both 76%) and in adherence to ART (51%) can be found during incarceration, with very lower percentages pre-incarceration and post-release; also, from the very same data, the highest percentage of undetectable viremia is located during incarceration (40%) rather than before it or post release, at which only 21% show undetectable viremia [32]. However, the high rate of turnover represents a challenge in assuring HIV care continuity to inmates [2].

Recent drug use and elevated level of addiction are extremely common among HIV prisoners as they enter jail, and this is due to both demographic and social instability characteristics [70]. Drug use is one of the main factors responsible for a marked reduction in the engagement of seropositive detainees in HIV treatment retention in care [70,74]. Moreover, among HIV positive inmates who use drugs and are on antiretroviral therapy, alcohol use and incarceration in the 6 months prior to beginning antiretroviral therapy are negatively associated with achieving undetectable pVL [75]. Moreover, according to a study by Palepu et al. [75], an HIV-positive inmate with a history of detention within 12 months of beginning ART has a higher probability of non-adherence; therefore, a lower probability of achieving undetectable pVL. However, there is some evidence on the fact that the longer the condemnation, the higher the probability of reaching undetectable viremia [76]. 

Nonetheless, the strategies to ensure the continuation of HIV/AIDS care for inmates living with HIV must be a public health priority [76].

## 6. After Release: Does the Situation Really Get Better?

HIV and retention in care pose a problem not only in correctional facilities, but also after being released from jails. The major issue is represented by adherence to ART.

Anxiety about release is a cause of low adherence to ART and poor retention in care. Linkage to care drops from 76% to 36% after release and retention in care from 76% to 30%, according to HIV care cascade [32]. Moreover, receipt of ART drops from 51% to 29% after release, affecting the percentage of undetectable viremia, which reaches 21% after release, from 40% during incarceration [32].

Adherence is related to “life chaos”, defined as “the perceived impossibility of making plans” in a study by Takada et al. [6]. This study included 356 HIV-positive individuals released from jail, and “life chaos” was found to be associated with low adherence to ART, poor retention in care, and, therefore, with low rates of access to the cures [6]. Moreover, this study found that “life chaos” was more common in those individuals who were infected during their jail time, and it especially burdened ethnic minorities, such as Afro-Americans and Hispanics [6]. However, the results of this study might have been influenced by the fact that these ethnic minorities represent the most frequent ethnicities among inmates in the US. Goodman-Meza et al. [1] have also shown that upon release inmates return to high-risk behaviors such as condomless sex, sex with multiple partners, chemsex, and injective drug use. 

The effects of detention on HIV infection spread are not only seen during the period in jail, but also after release. Important issues in the after-release period are hardships in providing housing and a job, the ever-present stigma towards HIV, previous imprisonment, the continuity in therapy, and the lack of a healthcare net, which often leads to poor adherence to treatment [77,78,79]. Although, as already stated, in a study by Eastment et al. [73], 62% of people had undetectable pVL within a year of release, other studies prove the difficulties of retention in care after release. In a study by Tiruneh et al., 43% of the included ex-inmates failed to connect to community HIV care [80]. An especially critical issue is the fact that most people at release are not given a 30 day supply of therapy, which leads to discontinuation in many cases [81]. The most vulnerable communities in this sense are the black people and PoC in general, MSM, transgender people, women, and people with psychiatric comorbidities [82,83]. Moreover, HIV-positive ex-inmates are not only affected by an increased risk of failing to reach or maintaining undetectable pVL, but also by an increased mortality. Primary causes of this increased mortality are AIDS in 46% of people, as a consequence of virological failure, followed by drug overdose in 15% of the ex-inmates, liver disease, cardiovascular disease, and accidental injury or suicide [84]. Therefore, despite the reduction in some risky behaviors mostly occurring during the period in jail, other major issues prevent PLWH from an optimal adherence and people with an unknown serostatus from accessing HIV testing [85]. 

Inmates need a systematic approach to emergency, routine, and preventive care in jail to create a healthier correctional environment, and a healthier community after release [4]. They should be referred to community organizations for the treatment of drug use disorders and the follow-up of medical care before they are released from detention facilities or correctional institutes [4]. Furthermore, because of the fluid nature of the jail population, the preventive and cure interventions have an impact not only in the detention facilities, but also in the communities to which they return [86].

## 7. Future Perspectives

An essential way to prevent transmission is to educate inmates about the importance of condoms and make condoms more accessible, too. In fact, unprotected sex is the main route of transmission among inmates [87]. An interesting option is the possibility of making PrEP more accessible among inmates. As a matter of fact, there are many movements in the US promoting the diffusion of PrEP among detainees, although it should include education programs about prophylaxis [19,88,89].

Although incarceration often represents a disruption in retention in care and viral suppression of PLWH, some authors highlight the opportunity to re-engage seropositive people in care by using new strategies. An interesting strategy is the Data-to-Care (D2C) approach, an approach that uses HIV surveillance data, pharmacy fill data, clinic appointment data, and other treatment to improve continuity of care in PLWH in the United States [90]. It operates in and out of correctional facilities and it aims to identify PLWH not in care or who are not viro-suppressed, and to guarantee appropriate medical and social care [90].

A project meant to guarantee the continuity in adherence to ARV for PLWH in Los Angeles, not only in jail but especially after release, is the Linking Inmates to Care in Los Angeles (LINK LA), a peer navigation intervention for HIV-infected men and transgender women released from jail. It starts while enrolled participants are still in jail and, through one meeting with a counselor every two weeks, it leads to consultations and problem-solving in HIV care and adherence; it also promotes activities and positive health expectations [91]. This, along other initiatives in HIV continuity of care after release, and the rates of viral suppression, demonstrate that long-term care for detainees helps control the spread of HIV and its consequences [92,93,94,95,96].

## 8. Conclusions

Infectious diseases are proven to be an important issue for people living in correctional facilities, mostly because of the crowded detention system. 

Among the infectious diseases, sexually transmitted infections are the most prevalent, and HIV is one of the most concerning problem in jails. 

Scarce knowledge on HIV and its transmission is the main reason behind the spread of the virus among inmates. Risky behaviors, such as condomless sex and sexual violence among detainees, are factors that contribute to the virus spread. 

The period after release from jail is no better in terms of compliance with ART.

Immediate interventions need to be made to face HIV diffusion among detainees, to improve the life conditions of PLWH in prison, and to keep track of them after release. Investments in prophylaxis such as PrEP and means to improve knowledge of inmates on HIV could slow and, hopefully, stop the virus diffusion in jail.

## Figures and Tables

**Table 1 healthcare-10-02380-t001:** Risk factors for HIV transmission and challenges for diagnosis and retention in care.

Risk Factors for HIV Transmission	Challenges for Diagnosis/Retention in Care
Scarce knowledge on HIV	Lack of screening programs
Unprotected sex	Mental disorders and scarce adherence to ART
Injection drug use	
Scarce investment on prophylaxis	
Partner violence	

## Data Availability

Not applicable.

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
