# Peer review of "Diagnosis, Treatment, and Prevention of HIV Infection among Detainees: A Review of the Literature"

_healthcare, 2022, doi:10.3390/healthcare10122380_

Round 1

Reviewer 1 Report

I think the work is interesting with innovative aspects and deserves publication. However, I want to suggest some improvements that can be made.

- Authors could be more specific about materials and methods in particular about search and exclusion criteria. Which items have you excluded and why?

- You could mention the Italian situation of penitentiary medicine example cit: doi: 10.3390 / healthcare9111511.

- in the section dedicated to COVID you could mention the need for an equal access to vaccines: doi: 10.3390 / vaccines9060538.

Author Response

I think the work is interesting with innovative aspects and deserves publication. However, I want to suggest some improvements that can be made.

- Authors could be more specific about materials and methods in particular about search and exclusion criteria. Which items have you excluded and why?

Response: we explained better materials and methods and clarified the exclusion criteria.

- You could mention the Italian situation of penitentiary medicine example cit: doi: 10.3390 / healthcare9111511.

Response: thank you for the suggestion, we added a few lines and the reference as well.

- in the section dedicated to COVID you could mention the need for an equal access to vaccines: doi: 10.3390 / vaccines9060538.

Response: thank you for the suggestion, we added a few lines and the reference as well.

Reviewer 2 Report

The research deals with the important topic of HIV infection among detainees. Although it is highly commendable that the conducted research deals with this argument, there are some serious shortcomings, and I hope that my comments will help to eliminate them and contribute to the quality of the work

Introduction

Lines 41-43: I am not surprised that half of the new HIV infections are diagnosed in black people, when they represent 42% of the jail population, but perhaps this is not the concept that the authors wanted to convey. I therefore ask the authors to make their idea more explicit

Lines 43-45. Wali et al…showed that HIV infection was only found in 4% of the included people. What is the rate of infection in the general population in Pakistan?

Lines 51-52 “The high prevalence of substance abuse, and in particular methamphetamine, leads to a decreased rate of HIV-RNA plasma viral load (pVL) suppression” it is not the high prevalence of substance abuse that leads to a decreased rate, but the substances abuse itself. Moreover, reference 5 does not support this statement, nor 10 nor 11.

the introduction is overall approximate and not very focused on the topic of the review

Risk factors and challenges for retention in care

I am not sure that many of the aspects considered in this chapter be really “risk factors”

lines  83-84 “In fact, a high percentage of HIV infections in jail involves Men who have Sex with Men (MSM)” Can you quantify "high"? Are there other studies that support this finding?

Lines 97-98 “a survey by Meyer et al [25] showed that among 84 HIV-positive participants, 49% suffered from sexual abuse” I suggest that the authors introduce this reference better. if the reference is about transgender inmates, perhaps the authors could better contextualize and provide more details to facilitate understanding

Line 98 “Female inmates are significantly more burdened by sexual abuse than male ones” are authors talking about sexual abuse during retention, or before entering prison? I think it is appropriate to keep the two aspects separate

Infectious comorbidities

the part relating to sexually transmitted and infectious diseases is very extensive and perhaps disproportionate to the rest of the contents, if you think you can attribute such importance to the topic, insert it in the title and in the discussion, otherwise I ask that it be significantly reduced. Moreover this part contains, in my opinion, inappropriate self-citations by authors

Line 139 “can facilitate HIV infection” it’s probably more adequate to talk about HIV transmission

Line 209 “To limit the transmission of HIV and TB infections among detainees the application of screening protocols at the arrest” perhaps the verb is missing

Lines 229-230: “New drugs administration routes, such as injective antiretroviral drugs, will be useful to promote the adherence even if similar situations will rise in the future, especially in detention settings” Could you please help me understand why injecting drug administration should help adherence to therapy in the event of an upcoming covid-like situation, when it was just said that due to the pandemic “clinicians worldwide were reallocated to address the emergency, leaving PLWH with an uncertain access to treatment and cures”?

ART in jail

Lines 236-238: “Unfortunately, the most vulnerable populations, such as homeless people, do not have a facilitated access to healthcare; therefore, their compliance to ART 237 is suboptimal [69].” I believe that if the authors want to introduce this reference, they should explain the link between homeless and ART in jail, because, as it is now, it is not immediate

Lines 70-72 “Eastment et al [70] showed that alt-242 hough 51% of the HIV-positive inmates had a detectable pVL upon their arrest, 62% of 243 them had still an undetectable pVL within a year of release” this statement is probably more adequate within the chapter “6. After release: does the situation really get better?”

After release: does the situation really get better?

“Anxiety about release is a cause of low adherence to ART and poor retention in care” is anxiety about release perceived before or after release? the paragraph is very interesting and certainly deals with very important topics, but I find some confusion between what happens before the release, when the inmate can still take advantage of the treatment that the prison provides, and after the release, when he is autonomous in its management

Future perspectives

please organize initiatives to separate those that can be implemented during detention, those that continue beyond detention and those that must be implemented after release

Author Response

The research deals with the important topic of HIV infection among detainees. Although it is highly commendable that the conducted research deals with this argument, there are some serious shortcomings, and I hope that my comments will help to eliminate them and contribute to the quality of the work

 Response: thank you for your work and your comments, we tried our best to address every one of them. We hope the paper meets your standards.

Introduction

Lines 41-43: I am not surprised that half of the new HIV infections are diagnosed in black people, when they represent 42% of the jail population, but perhaps this is not the concept that the authors wanted to convey. I therefore ask the authors to make their idea more explicit

Response: we rephrased that line.

Lines 43-45. Wali et al…showed that HIV infection was only found in 4% of the included people. What is the rate of infection in the general population in Pakistan?

Response: we revised the introductions, so we removed that line. However, Pakistan is estimated to have  25,000 new HIV infections per annum. The estimated population of Pakistan is 162,419,946. The general population prevalence of HIV is estimated as 0.1% and high-risk population prevalence as 1-2%.

Lines 51-52 “The high prevalence of substance abuse, and in particular methamphetamine, leads to a decreased rate of HIV-RNA plasma viral load (pVL) suppression” it is not the high prevalence of substance abuse that leads to a decreased rate, but the substances abuse itself. Moreover, reference 5 does not support this statement, nor 10 nor 11.

Response: we corrected the sentence. We removed the reference 5, as for references 10 and 11 (now 11 and 12), they regard the high prevalence of HIV among drug users, so we moved those reference and we introduced them differently.

the introduction is overall approximate and not very focused on the topic of the review

Response: we revised the introduction.

Risk factors and challenges for retention in care

I am not sure that many of the aspects considered in this chapter be really “risk factors”

Response: what we believe is the main risk factor is the inadequate knowledge on HIV, to which all the other behaviors, such as condomless sex or chemsex, may be consequential. All the other aspects enumerated in the section are easily what we consider risk factors, as unprotected sex, violent sexual intercourse or drug abuse, or challenges for retention in care, as per title.

lines  83-84 “In fact, a high percentage of HIV infections in jail involves Men who have Sex with Men (MSM)” Can you quantify "high"? Are there other studies that support this finding?

Response: we clarified in the paper and added more evidences.

Lines 97-98 “a survey by Meyer et al [25] showed that among 84 HIV-positive participants, 49% suffered from sexual abuse” I suggest that the authors introduce this reference better. if the reference is about transgender inmates, perhaps the authors could better contextualize and provide more details to facilitate understanding

 Response: the reference is about inmates in general, but specifically women. We clarified in the paper.

Line 98 “Female inmates are significantly more burdened by sexual abuse than male ones” are authors talking about sexual abuse during retention, or before entering prison? I think it is appropriate to keep the two aspects separate

Response: we clarified in the manuscript.

Infectious comorbidities

the part relating to sexually transmitted and infectious diseases is very extensive and perhaps disproportionate to the rest of the contents, if you think you can attribute such importance to the topic, insert it in the title and in the discussion, otherwise I ask that it be significantly reduced. Moreover this part contains, in my opinion, inappropriate self-citations by authors

Response: we reduced the part of the sexually transmitted infections. Despite HIV remains the main topic of the review, we believe that it is important to consider the other STIs. Also, we revised the citations.

Line 139 “can facilitate HIV infection” it’s probably more adequate to talk about HIV transmission

Response: we corrected that line.

Line 209 “To limit the transmission of HIV and TB infections among detainees the application of screening protocols at the arrest” perhaps the verb is missing

Response: we fixed the line.

Lines 229-230: “New drugs administration routes, such as injective antiretroviral drugs, will be useful to promote the adherence even if similar situations will rise in the future, especially in detention settings” Could you please help me understand why injecting drug administration should help adherence to therapy in the event of an upcoming covid-like situation, when it was just said that due to the pandemic “clinicians worldwide were reallocated to address the emergency, leaving PLWH with an uncertain access to treatment and cures”?

 Response: what lacked during the pandemic was the contact between the patients and their clinicians, so the patients were left without the regular appointments to check their viroimmunological status and their well being as well. In this situation many people may not have an optimal adherence to ART. Although, with injective antivirals the appointment every two months (as it is at the moment for long acting) must be guaranteed, even during emergency, and the administration of ART is no longer left on the patient’s hands. Also, it helps keeping that aforementioned contact with the doctor. Of course this is not applicable to every situation, but what we aimed to say is that some categories of patients could have benefited by long acting antivirals during lockdown.

ART in jail

Lines 236-238: “Unfortunately, the most vulnerable populations, such as homeless people, do not have a facilitated access to healthcare; therefore, their compliance to ART 237 is suboptimal [69].” I believe that if the authors want to introduce this reference, they should explain the link between homeless and ART in jail, because, as it is now, it is not immediate

Response: we clarified that part in the manuscript.

Lines 70-72 “Eastment et al [70] showed that alt-242 hough 51% of the HIV-positive inmates had a detectable pVL upon their arrest, 62% of 243 them had still an undetectable pVL within a year of release” this statement is probably more adequate within the chapter “6. After release: does the situation really get better?”

Response: we added a line referring to it in paragraph 6, although we did not entirely move that sentence because we believe it is important in demonstrating the efficacy of ART initiated in prison.

After release: does the situation really get better?

“Anxiety about release is a cause of low adherence to ART and poor retention in care” is anxiety about release perceived before or after release? the paragraph is very interesting and certainly deals with very important topics, but I find some confusion between what happens before the release, when the inmate can still take advantage of the treatment that the prison provides, and after the release, when he is autonomous in its management

Response: the anxiety may be perceived before the release but it is caused by the release itself and the period after it. As states in the paper, in many situations ex-inmates face difficulties in find jobs and house, after being released. This often leads to poor adherence to ART.

Future perspectives

please organize initiatives to separate those that can be implemented during detention, those that continue beyond detention and those that must be implemented after release

Response: we reorganized that part.

Reviewer 3 Report

English language needs editing. Please check lines 51 and 52, may need clarification or correction.

Is there information on prep availability in jails? Also is there information on the prevalence on HIV positive persons upon admission to jail and new cases while in jail. This would be very interesting. Also to make distinction about MSM upon arrival and those that were sexually abused and are now considered to be MSM.

More detail should be provided for the screening programs mentioned in the Manuscript.

Can you please present risk factors as a figure or table?

Please write conclusions in a more concise manner.

Author Response

English language needs editing. Please check lines 51 and 52, may need clarification or correction.

Response: we revised English and those lines in particular.

Is there information on prep availability in jails? Also is there information on the prevalence on HIV positive persons upon admission to jail and new cases while in jail. This would be very interesting. Also to make distinction about MSM upon arrival and those that were sexually abused and are now considered to be MSM.

Response: we did not find information on availability pf PrEP in jail. Although there are movements in USA, such as the RIGHT act, that promotes availability of PrEP in jail. We added data and a reference about the situation upon incarceration and after release, in terms of diagnosis, ART and viremia.

More detail should be provided for the screening programs mentioned in the Manuscript.

Response: we provided more details about the screening programs.

Can you please present risk factors as a figure or table?

Response: we added a table with major risk factors, as suggested.

Please write conclusions in a more concise manner.

Response: we revised the conclusions as suggested.

Reviewer 4 Report

The presented review is aimed to investigate the conditions that favor HIV spread among detainees, the issues People Living With HIV (PLWH) have to face while being in jail and the difficulties in maintaining compliance to ART after release. Material and Methods are clearly described and sufficiently explained. The authors concluded that among the infectious diseases, sexually transmitted infections are the most prevalent and HIV is one of the most concerning problem in jails. Risky behavior, like condomless sex, lack of attention for other STIs and sexual violence among detainees are the factors that contribute to the virus spread. The period after release from jail has proved to be no better in terms of compliance to ART, due to difficult living situations for ex-detainees and the difficulties in getting antiretroviral drugs. Some categories of people seem to be mostly affected by these problems, like Hispanics, black people, women, trans-gender and MSM. The material is very interesting and significant. The English language is appropriate and understandable.

Author Response

The presented review is aimed to investigate the conditions that favor HIV spread among detainees, the issues People Living With HIV (PLWH) have to face while being in jail and the difficulties in maintaining compliance to ART after release. Material and Methods are clearly described and sufficiently explained. The authors concluded that among the infectious diseases, sexually transmitted infections are the most prevalent and HIV is one of the most concerning problem in jails. Risky behavior, like condomless sex, lack of attention for other STIs and sexual violence among detainees are the factors that contribute to the virus spread. The period after release from jail has proved to be no better in terms of compliance to ART, due to difficult living situations for ex-detainees and the difficulties in getting antiretroviral drugs. Some categories of people seem to be mostly affected by these problems, like Hispanics, black people, women, trans-gender and MSM. The material is very interesting and significant. The English language is appropriate and understandable.

Response: thank you very much for you work and for your words, they are very appreciated. We are glad that you agree with us on the importance of the topic.

Round 2

Reviewer 2 Report

the authors have answered all my questions, but I point out that there is a problem in the numbering of the references: on line 557 there is a reference "number 1" and 4 others are from line 608 onwards. it is not clear where these references are introduced in the text, since the numbering does not match

Author Response

We checked the lines you mentioned and all the other references in the manuscript, but we could not find mismatches in them or other mistake. At line 557 there is reference n° 75 which is mentioned in the text at line 266. At line 608 and onward there are references from 93 to 96, that are mentioned at line 341 in the text. If it is possible we would like to be pointed out the mistake a little bit more precisely, so we can correct it. Thank you again for your work.